# Caregivers' practices and factors associated with malaria vaccine uptake among under-five children in the Tiko Health District, Cameroon: A community based cross-sectional study

Idang Maureen Abiache[1], Divine Nsobinenyui[2], Chrisantus Eweh Ukah[3]*,
Yunika Larissa Kumenyuy[3], Ngu Claudia Ngeha[3], Randolf Wefuan[3],
Syveline Zuh Dang[3], Ndip Esther Ndip[3], Mirabelle Pandong Feguem[3],
Dickson S. Nsagha[3]

**1** Faculty of Medicine and Pharmaceutical Sciences, Saint Monica University Higher Institute of Buea, Buea, Cameroon, **2** Department of Applied Zoology, Faculty of Science, University of Bamenda, Bamenda, Cameroon, **3** Department of Public Health and Hygiene, Faculty of Health Sciences, University of Buea, Buea, Cameroon

* chrisantuseweh@gmail.com

## Abstract

Malaria remains a leading cause of morbidity and mortality among children under five in Cameroon. In 2021, the World Health Organization (WHO) recommended the RTS,S/AS01 malaria vaccine for children in areas with moderate to high transmission. This study assessed caregivers' practices and factors associated with malaria vaccine uptake among under-five children in the Tiko Health District. A community-based cross-sectional study was conducted from March to April 2025, involving 410 caregivers of children aged 0–5 years. Participants were selected through multistage sampling. Data were collected using a structured questionnaire and analyzed with descriptive statistics and logistic regression to identify factors associated with vaccine uptake.. Variables with p < 0.20 in bivariate analysis were included in the multivariable model, and adjusted odds ratios(aOR) with 95% confidence intervals(CI) were reported. Only 32.2%(n = 132) of children had received the malaria vaccine. Of those vaccinated, 72.0% completed the recommended doses, and 82.6% of caregivers maintained vaccination records. Multivariable analysis revealed that children of female caregivers(aOR: 4.16, 95% CI: 1.47–11.75), caregivers in health professions(aOR: 2.87, 95% CI: 1.35–5.69), biological parents(aOR: 11.44, 95% CI: 1.52–86.11), and those with household income of 89USD–179USD(aOR: 2.76, 95% CI: 1.68–4.55) had significantly higher odds of vaccine uptake. Trust in health workers(aOR: 6.12, 95% CI: 2.97–12.61) and information from healthcare providers(aOR: 7.60, 95% CI: 3.82–15.08) were also strong predictors. Conversely, prior malaria infection in children was associated with lower odds of vaccination(aOR: 0.31, 95%CI: 0.18–0.54). Malaria vaccine uptake among under-five

**Data availability statement:** All relevant data underlying the findings of this study are provided within the manuscript and its Supporting Information files. The dataset has been fully anonymized to protect participant confidentiality.

**Funding:** The authors received no specific funding for this work.

**Competing interests:** The authors have declared that no competing interests exist.

children in the Tiko Health District is suboptimal. Caregiver sex, profession, household income, and access to trusted health information significantly influenced uptake. Strengthening caregiver education, improving healthcare access, and enhancing trust in health providers are vital to increase malaria vaccine coverage.

## Background

Malaria continues to be a public health concern in sub-Saharan Africa, particularly affecting children under five years of age. According to the World Health Organization [1], there were approximately 263 million malaria cases and 597,000 malaria-related deaths globally in 2024, with over 94% of the cases and 95% of the deaths occurring in the WHO Africa Region. Young children account for more than 76% of all malaria-related fatalities (WHO, 2024) [1]. In Cameroon, malaria is a leading cause of morbidity and mortality among young children. The country reports over seven million malaria cases annually, with children under five being the most affected group [2,3]. The introduction of the RTS, S/AS01 malaria vaccine, known as Mosquirix, represents a significant advancement in malaria prevention. In October 2021, the WHO recommended the widespread use of this vaccine for children in regions with moderate to high transmission of *Plasmodium falciparum* (WHO, 2021) [4]. Studies from pilot countries conducted in Ghana, Kenya, and Malawi have shown a 30% reduction in severe malaria cases and hospitalizations among children who completed the four-dose series [5]. Cameroon became one of the first countries outside the pilot programs to integrate the malaria vaccine into its routine immunization plan, launching the program in January 2024 [6]. Despite these advances, the success of the malaria vaccine program depends on achieving high coverage and acceptance within communities. Socioeconomic status, cultural norms, and exposure to misinformation also influence decision-making regarding malaria vaccination [7]. Vaccine hesitancy is also a significant factor contributing to the low vaccination coverage observed in many African countries [8]. Furthermore, lack of awareness, financial challenges, cultural beliefs and logistical challenges pose a problem to malaria prevention [9]. Understanding caregiver practices and the factors influencing vaccine uptake is important for the success of the vaccination program. Studies have shown that caregiver education level, attendance at antenatal clinics, and closeness to vaccination centers significantly affect vaccine [10]. In Cameroon, the Ministry of Public Health, in collaboration with global partners are working on malaria vaccine program as part of its integrated malaria control strategy. Parents and caregivers are very important in making decisions about their children's healthcare, particularly in ensuring vaccination uptake and following recommended immunization schedules [11,12]. Caregiver practices, awareness of vaccine availability, and health-seeking behaviors, are important to ensuring complete and timely vaccination. Therefore, this study, aimed to assess the uptake of the malaria vaccine among under-five children in the Tiko Health District. Specifically, it seeks to assess caregivers' practices toward malaria vaccination and identify factors influencing vaccine uptake.

## Methodology

### Study design

This was a community-based cross-sectional design aimed at assessing caregivers' practices and factors associated with uptake of the malaria vaccine and its determinants among children aged 0–5 years in the Tiko Health District of the South West Region of Cameroon.

### Study area

The study was conducted in Tiko Health District (THD), located in Fako Division, South West Region of Cameroon, from the 1st of March 2025 to the 11th of April 2025. Tiko is comprised of eight health areas and serves an estimated population of 55,914 people with diverse ethnic and socioeconomic backgrounds. The health district is characterized by a combination of urban and peri-urban settlements, and it presents unique public health challenges such as poor sanitation and limited access to health services.

### Study population

The target population consisted of caregivers of children aged 0–5 years residing in the Tiko Health District during the study period. For this study, 'caregivers' were defined as adults (≥18 years) primarily responsible for the day-to-day care of the child, including mothers, fathers, or other guardians. Caregivers were eligible to participate if they were the main provider of care for an under-five child residing in the Tiko Health District

### Sample size determination

The sample size was calculated using the formula for estimating a population proportion with a 95% confidence level [13,14]:

$$n = \frac{Z^2 p(1-p)}{e^2}$$

Where;
Z= 1.96 (for 95% confidence level)
P = 0.5 (estimated proportion). The p was chosen as 50% (0.5) because the malaria vaccine was approved in Cameroon not along ago and data on the uptake is not readily available.
e= 0.05 (margin of error)
This gave us a sample size of 384, which was increased to 410 to account for non-response rate.

### Sampling technique

A multistage sampling technique incorporating probability proportional to size (PPS) was used to recruit under-five children for this study. In the first stage, simple random sampling by balloting was used to select six out of the eight health areas in the Tiko Health District. Each health area name (Holforth, Mutengene, Likomba, Mondoni, Misselele, Mudeka, Kange, and Tiko Town) was written on separate slips of paper, folded, placed in a container, and six were randomly drawn without replacement. The selected health areas were: Likomba, Holforth, Mutengene, Misselele, Tiko Town, and Kange, representing a mix of urban and rural communities.

In the second stage, probability proportional to size (PPS) sampling was used to determine the number of participants to be recruited from each selected health area based on updated under-five population estimates obtained from the Tiko District Health Services. The total under-five population across the six selected health areas was 20,641, and the overall sample size was 410 children. The number of children recruited from each health area was proportional to the size of the health area as shown on Table 1:

**Table 1. Number of participants selected per health area.**

| Health area | Under-five population | % of Total U5 Pop. | Allocated sample (n) |
|---|---|---|---|
| Holforth | 5,706 | 27.60% | 113 |
| Mutengene | 8,170 | 39.60% | 162 |
| Likomba | 1,798 | 8.70% | 36 |
| Misselele | 955 | 4.60% | 19 |
| Kange | 709 | 3.40% | 14 |
| Tiko Town | 3,303 | 16.00% | 66 |
| **Total** | **20,641** | **100%** | **410** |

In the third stage, community health workers (CHWs) assisted in identifying and mapping households with at least one child under the age of five in each selected health area. These mapped households formed the sampling frame for each area. Field workers approached households consecutively and enrolled one eligible under-five child per household. If a household had more than one eligible child, simple random sampling by balloting was used to select one child. Recruitment in each health area continued until the allocated sample size for that area was achieved.

This approach ensured a representative and proportionate sample across the selected health areas, while also leveraging local knowledge from CHWs to identify and access eligible households in the absence of a pre-existing household register.

## Inclusion and exclusion criteria

Caregivers of under-five children who gave their consent were included in the study while those who were severely ill were excluded.

## Data collection tools and procedures

A structured pre-tested questionnaire was developed following a review of relevant literature on malaria vaccination and childhood immunization practices in sub-Saharan Africa [15–17]. Items were adapted from previously validated survey instruments, modified to reflect the study context, and reviewed by two experts in public health research for content validity. The questionnaire was pretested in a neighboring community (Mutengene Health Area) to ensure clarity, cultural appropriateness, and reliability before being finalized for data collection. Necessary revisions were made before the full rollout.

The questionnaire was divided into four sections: Section A: Socio-demographic information, Section B: Knowledge about the malaria vaccine, Section C: Attitudes towards the malaria vaccine and Section D: Use and practices regarding malaria vaccination.

Data were collected by the principal investigator and four trained research assistants who were public health graduates with prior experience in community-based surveys. Before data collection, they underwent a two-day training facilitated by the principal investigator. The training covered the study objectives, ethical considerations, procedures for obtaining informed consent, confidentiality, interviewing techniques, and a pilot administration of the structured questionnaire. For literate participants, the questionnaire was self-administered, while illiterate participants were interviewed by research assistants who read questions aloud and recorded responses.

## Data management and analysis

Completed questionnaire was reviewed for completeness and accuracy, coded, and entered into SPSS version 26 for analysis. Descriptive statistics summarized demographic and practice-related variables. Practices were scored based on correct responses, and a 60% cutoff determined "good" vs. "poor" practices [18].

Bivariable logistic regression was conducted to assess associations between each independent variable and malaria vaccine uptake. Variables with a p-value less than 0.20 in the bivariate analysis were retained for the multivariable logistic regression model. The multivariable model was used to control for confounding and determine independent predictors of vaccine uptake. Adjusted odds ratios (aOR) with 95% confidence intervals (CI) and p-values were reported. For categorical variables, reference groups were clearly specified (e.g., "Male" for caregiver sex, "No formal education" for education level, and "89USD" for income). All costs were originally recorded in Central African Francs (XAF) and converted to United States Dollars (USD) using the prevailing exchange rate at the time of the study (1 USD = XAF559.18).

### Ethical considerations

Ethical clearance was obtained from the Southwest Ethics Committee for Human Health Research in Buea (135/CRERSH/SW/C/02/2025). Administrative authorization was secured from the Faculty of Health Sciences of the University of Douala, and the Tiko District Health Service. Participants were briefed on the study's purpose, benefits, and voluntary nature. Written informed consent was obtained from all participants. Confidentiality and anonymity were strictly maintained by using unique codes instead of names on questionnaire.

## Results

### Socio-demographic characteristics

The mean age of the 410 caregivers was 33.6 and the standard deviation was 8.9. A total of 219 (53.4%) of caregivers were within the age group 21–35 years and 341 (83.2%) of them were females. Secondary education was the dominant educational level 160 (39.0%) and 340 (82.9%) were Christians. A majority 249 (60.7%) were married and 158 (38.5%) were self-employed. A vast majority 329 (80.2%) of the caregivers were direct parents of the under-five children and 391 (95.4%) were non-smokers (Table 2).

### Practices of under-five caregivers toward the malaria vaccine

Regarding the caregivers' practices towards the malaria vaccine (Table 3), 278 (67.8%) children were vaccinated against malaria. Of the 132 whose children were vaccinated, 109 (82.6%) reported keeping vaccination records and 98 (74.2) reported following up when their children missed a vaccination schedule. A total of 60 (44.8%) of those who children had taken the vaccine had taken 3 doses (Fig 1).

Regarding the overall practices of caregivers of under-five children towards the malaria vaccine, they were asked a total of seven questions with a maximum obtainable score of 7. The cutoff point for good practices was 60%. Hence, those who scored from 4 and above were classified as having overall good practices and those who scored below 4 as having poor practices. The overall good practices were 24.1%.

### Factors associated with vaccine uptake among under-five children

Regarding factors associated with the malaria vaccine uptake (Table 4), ten factors were founded significantly associated with malaria vaccine uptake in the bivariable analysis using simple logistic regression analysis. These factors were sex, marital status, occupation, relationship with child, household income, accessibility of health services, previous history of malaria, source of information, trust in health workers, and going for general child vaccination.

### Factors independently associated with malaria vaccine uptake among under-five children

Regarding factors independently associated with the malaria vaccine uptake by under five children (Table 5), eight factors were found independently associated with the vaccine uptake. These were; sex, profession, relationship with child, income, accessibility of health services, history of malaria infection, source of information, and trust in health workers.

**Table 2. Socio-demographic characteristics of caregivers.**

| Variable | Category | Frequency | Percentage |
|---|---|---|---|
| Age group (years) | <21 | 20 | 4.8 |
| | 21-35 | 219 | 53.4 |
| | 36-50 | 149 | 36.3 |
| | 51-65 | 22 | 5.4 |
| | **Total** | **410** | **100** |
| Sex | Male | 69 | 16.8 |
| | Female | 341 | 83.2 |
| | **Total** | **410** | **100** |
| Education | No formal | 71 | 17.3 |
| | Primary | 81 | 19.8 |
| | Secondary | 160 | 39.0 |
| | Tertiary | 98 | 23.9 |
| | **Total** | **410** | **100** |
| Religion | Muslim | 70 | 17.1 |
| | Christian | 340 | 82.9 |
| | **Total** | **410** | **100** |
| Marital status | Single | 134 | 32.7 |
| | Married | 249 | 60.7 |
| | Widowed | 23 | 5.6 |
| | Divorced | 4 | 1.0 |
| | **Total** | **410** | **100** |
| Employment status | Student | 49 | 12.0 |
| | Unemployed | 83 | 20.2 |
| | Self-employed | 158 | 38.5 |
| | Employed | 120 | 29.3 |
| | **Total** | **410** | **100** |
| Relation with child | Not related | 22 | 5.4 |
| | Other relative | 36 | 8.8 |
| | Parents | 329 | 80.2 |
| | Sibling | 23 | 5.6 |
| | **Total** | **410** | **100** |
| Number of children | 1-2 | 236 | 57.6 |
| | 3-4 | 131 | 32.0 |
| | 4+ | 43 | 10.5 |
| | **Total** | **410** | **100** |
| Household income (USD) | <89 | 138 | 33.7 |
| | 89-179 | 206 | 50.2 |
| | 179+ | 66 | 16.1 |
| Smoking status | Smoker | 19 | 4.6 |
| | Non-smoker | 391 | 95.4 |
| | **Total** | **410** | **100** |
| Alcohol consumption | No | 127 | 31.0 |
| | Yes | 283 | 69.0 |
| | **Total** | **410** | **100** |

*(Continued)*

**Table 2.** (Continued)

| Variable | Category | Frequency | Percentage |
|---|---|---|---|
| Accessible health services | No | 64 | 15.6 |
| | Yes | 346 | 84.4 |
| | **Total** | **410** | **100** |
| Child ever had Malaria Episodes | No | 61 | 14.9 |
| | Yes | 349 | 85.1 |
| | **Total** | **410** | **100** |
| Trust health workers | No | 95 | 23.2 |
| | Yes | 315 | 76.8 |
| | **Total** | **410** | **100** |
| Go for general routine vaccination | No | 136 | 33.2 |
| | Yes | 274 | 66.8 |
| | **Total** | **410** | **100** |

**Table 3. Practices of under-five caregivers toward the malaria vaccine.**

| Variable | Category | Frequency | Percentage |
|---|---|---|---|
| Child is vaccinated | No | 278 | 67.8 |
| | Yes | 132 | 32.2 |
| | **Total** | **410** | **100** |
| Keep vaccination records | No | 23 | 17.4 |
| | Yes | 109 | 82.6 |
| | **Total** | **132** | **100** |
| Follow up when child missed vaccination schedule | No | 34 | 25.8 |
| | Yes | 98 | 74.2 |
| | **Total** | **132** | **100** |
| Regularly seek information about malaria vaccine from reliable sources | No | 78 | 59.1 |
| | Yes | 54 | 40.9 |
| | **Total** | **132** | **100** |
| Compliance of recommended vaccine doses by the child | No | 37 | 28.0 |
| | Yes | 95 | 72.0 |
| | **Total** | **132** | **100** |
| Caregiver actively participate in community vaccination programs that offer the malaria vaccine to children | No | 85 | 64.4 |
| | Yes | 47 | 35.6 |
| | **Total** | **132** | **100** |
| Seeks medical advice in case of negative symptoms after vaccination | No | 62 | 47.0 |
| | Yes | 70 | 53.0 |
| | **Total** | **132** | **100** |

Under five children of female caregivers were about four times more likely to be vaccinated against malaria (aOR:4.16, 95%CI:1.47-11.75; p = 0.007) as compared to children of male caregivers. Children of caregivers who were their direct parents were about eleven times more likely to be vaccinated aOR:11.44, 95%CI:1.52-86.11; p = 0.008) as compared to those whose caregivers were not directly related to them. Household income was generally positively associated with

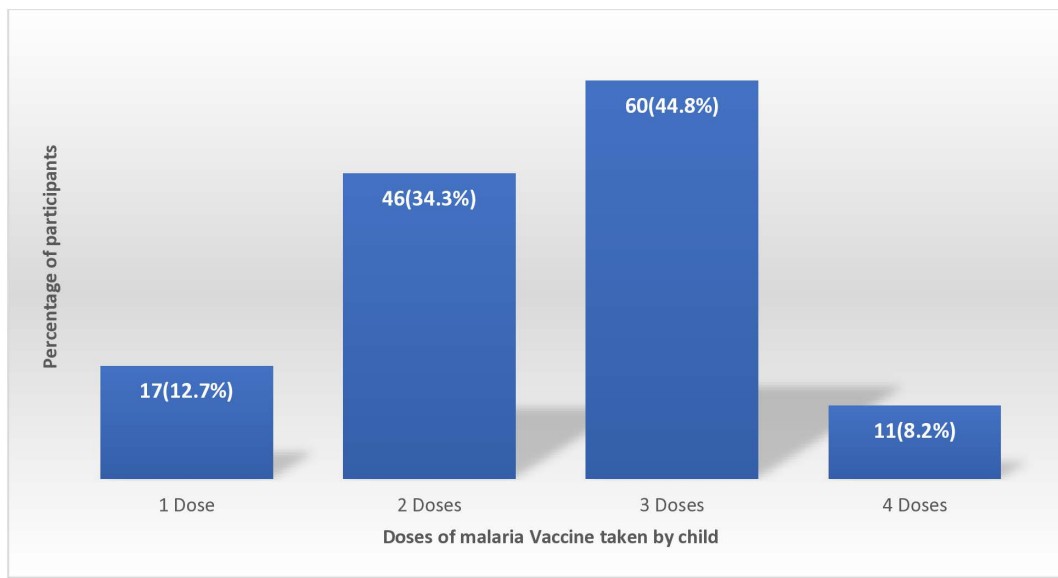

**Fig 1. Distribution of vaccinated children against malaria by number of doses taken.**

vaccine uptake with children whose caregiver average monthly income was between 89USD to 179USD being more than two times more likely to be vaccinated (aOR:2.76, 95%CI:1.68-4.55; p<0.001) as compared to children of caregivers whose average monthly income was less than 89USD. Children of caregivers who reported easily accessible health services were about two times more likely to be vaccinated (aOR:2.3, 95%CI:1.18-4.48; p=0.020) as compared to those who reported not having easily accessible health services. Caregivers who reported history of malaria infection among their children were 69% less likely to have their children vaccinated (aOR:0.31, 95%CI:0.18-0.54; p<0.001). Children of caregivers who main source of information on vaccination were more than seven times more likely to be vaccinated (aOR:7.60, 95%CI: 3.86-15.08, p<0.001) as compared to those whose main source of information was family/friends. Children of caregivers who reported trusting healthcare workers were about six times more likely to be vaccinated (aOR:6.12, 95%CI:2.97-12.61) as compared to those who reported not having trust in healthcare workers.

## Discussion

This study provides important insights into the uptake of the RTS,S/AS01 malaria vaccine among under-five children in the Tiko Health District, Cameroon, highlighting both caregiver practices and factors influencing vaccination among under-five children. Overall, malaria vaccine coverage remains suboptimal, with less than one-third of children receiving the vaccine. The findings underscore gaps in caregiver knowledge, health service utilization, and access to vaccination, which are critical for achieving the public health goal of reducing malaria morbidity and mortality in this vulnerable population. Understanding the characteristics of caregivers and household factors associated with vaccine uptake can inform targeted interventions to improve immunization coverage and strengthen malaria prevention strategies at the community level. The findings from this study provides important understanding into the uptake of malaria vaccine, its associated factors, and practices of under-five caregivers towards the malaria vaccine.

The study found that 32.2% of caregivers had vaccinated their under-five children against malaria, this indicates a low uptake of the malaria. These findings are similar to early uptake rates in pilot countries such as Ghana, Kenya, and Malawi, conducted in 2024, where initial community response to the malaria vaccine was low [19]. Among the 132 caregivers whose children received the malaria vaccine, a high proportion 82.6% reported keeping vaccination records,

**Table 4. Factors associated with vaccine uptake among under-five children using simple logistic regression.**

| Variable | Category | Child vaccinated | COR | 95% CI for COR | | p value |
|---|---|---|---|---|---|---|
| | | | | Lower | Upper | |
| Sex | Female | 117(28.5) | 1.88 | 1.017 | 3.475 | **0.044** |
| | Male | 15(3.7) | 1 | | | |
| Education | Tertiary | 40(9.8) | 1.76 | 0.91 | 3.39 | 0.091 |
| | Secondary | 50(12.2) | 1.16 | 0.63 | 2.15 | 0.638 |
| | Primary | 22(5.4) | 0.95 | 0.47 | 1.94 | 0.89 |
| | No formal | 20(4.9) | 1 | | | |
| Religion | Christian | 113(27.6) | 1.34 | 0.75 | 2.37 | 0.322 |
| | Muslim | 19(4.6) | 1 | | | |
| Marital status | Divorced | 1(0.2) | 1.06 | 0.11 | 10.57 | 0.959 |
| | Widowed | 7(1.7) | 1.39 | 0.53 | 3.69 | 0.503 |
| | Married | 92(22.4) | 1.87 | 1.16 | 3 | **0.010** |
| | Single | 32(7.8) | 1 | | | |
| Occupation | Employed | 39(9.5) | 1.88 | 0.85 | 4.15 | 0.119 |
| | Self-employed | 58(14.1) | 2.26 | 1.05 | 4.87 | **0.037** |
| | Unemployed | 25(6.1) | 1.68 | 0.73 | 3.89 | 0.225 |
| | Student | 10(2.4) | 1 | | | |
| Health related profession | Yes | 18(4.4) | 2.77 | 1.35 | 5.69 | **0.006** |
| | No | 114(27.8) | 1 | | | |
| Relationship to child | Siblings | 4(1.0) | 4.42 | 0.45 | 43.11 | 0.201 |
| | Parents | 116(28.3) | 11.44 | 1.52 | 86.11 | **0.018** |
| | Other relative | 11(2.7) | 9.24 | 1.1 | 77.58 | **0.041** |
| | Not directly related | 1(0.2) | 1 | | | |
| Income (USD) | 179 | 19(4.6) | 1.59 | 0.81 | 3.12 | 0.179 |
| | 89-179 | 85(20.0) | 2.76 | 1.68 | 4.55 | **<0.001** |
| | <89 | 28(6.8) | 1 | | | |
| Smoking status | Not smoker | 129(31.5) | 2.63 | 0.75 | 9.17 | 0.13 |
| | Smoker | 3(0.7) | 1 | | | |
| Alcohol consumption | Yes | 90(22.00 | 0.94 | 0.6 | 1.47 | 0.799 |
| | No | 42(10.2) | 1 | | | |
| Accessible health services | Yes | 120(29.3) | 2.3 | 1.18 | 4.48 | **0.014** |
| | No | 12(2.9) | 1 | | | |
| History of malaria in child | Yes | 98(23.9) | 0.31 | 0.18 | 0.54 | **<0.001** |
| | No | 34(8.3) | 1 | | | |
| Source of information on vaccines | Social media | 15(3.7) | 1.33 | 0.58 | 3.08 | 0.501 |
| | Others | 3(0.7) | 10.93 | 1.05 | 113.36 | **0.045** |
| | None | 2(0.5) | 0.05 | 0.01 | 0.24 | **<0.001** |
| | Health care provider | 98(23.9) | 7.6 | 3.82 | 15.08 | **<0.001** |
| | Family and friends | 14(3.4) | 1 | | | **<0.001** |
| Trust health workers | Yes | 123(30.0) | 6.12 | 2.97 | 12.61 | **<0.001** |
| | No | 9(2.2) | 1 | | | **<0.001** |
| Undergoes standard immunization | Yes | 116(28.3) | 5.51 | 3.1 | 9.78 | **<0.001** |
| | No | 16(3.9) | 1 | | | |

COR = Crude Odd Ratio.

**Table 5. Factors independently associated with malaria vaccine uptake among under-five children.**

| Variable | Category | Child vaccinated | AOR | 95% CI for AOR | | p value |
|---|---|---|---|---|---|---|
| | | | | Lower | Upper | |
| Sex | Female | 117(28.5) | 4.16 | 1.47 | 11.75 | **0.007** |
| | Male | 15(3.7) | 1 | | | |
| Health related profession | Yes | 18(4.4) | 2.87 | 1.35 | 5.69 | **0.008** |
| | No | 114(27.8) | 1 | | | |
| Relationship to child | Siblings | 4(1.0) | 4.42 | 0.45 | 43.11 | 0.201 |
| | Parents | 116(28.3) | 11.44 | 1.52 | 86.11 | **0.008** |
| | Other relative | 11(2.7) | 9.24 | 1.1 | 77.58 | **0.041** |
| | Not directly related | 1(0.2) | 1 | | | |
| Income (USD) | 179+ | 19(4.6) | 1.59 | 0.81 | 3.12 | 0.179 |
| | 89-179 | 85(20.0) | 2.76 | 1.68 | 4.55 | **<0.001** |
| | <89 | 28(6.8) | 1 | | | |
| Smoking status | Not smoker | 129(31.5) | 2.63 | 0.75 | 9.17 | 0.13 |
| | Smoker | 3(0.7) | 1 | | | |
| Drink alcohol | Yes | 90(22.00 | 0.94 | 0.6 | 1.47 | 0.799 |
| | No | 42(10.2) | 1 | | | |
| Accessible health services | Yes | 120(29.3) | 2.3 | 1.18 | 4.48 | **0.02** |
| | No | 12(2.9) | 1 | | | |
| Child has suffered from malaria before | Yes | 98(23.9) | 0.31 | 0.18 | 0.54 | **<0.001** |
| | No | 34(8.3) | 1 | | | |
| Source of information on vaccines | Social media | 15(3.7) | 1.33 | 0.58 | 3.08 | 0.501 |
| | Others | 3(0.7) | 10.93 | 1.05 | 113.36 | **0.045** |
| | None | 2(0.5) | 0.05 | 0.01 | 0.24 | **<0.001** |
| | Health care provider | 98(23.9) | 7.6 | 3.82 | 15.08 | **<0.001** |
| | Family and friends | 14(3.4) | 1 | | | |
| Trust health workers | Yes | 123(30.0) | 6.12 | 2.97 | 12.61 | **<0.001** |
| | No | 9(2.2) | 1 | | | **<0.001** |
| Go for general vaccination | Yes | 116(28.3) | 5.51 | 3.1 | 9.78 | **<0.001** |
| | No | 16(3.9) | 1 | | | |

AOR = Adjusted Odd Ratio.

which is an encouraging level of health responsibility. Similarly, 74.2% followed up when their child missed a vaccination appointment. These practices are important for ensuring full adherence to the vaccine, especially considering that RTS,S requires a multi-dose for optimal effectiveness [20]. Only 44.8% of vaccinated children had received at least three doses, suggesting challenges with vaccine completion, these results align with findings from Zambia and Tanzania, where multi-dose vaccine schedules were difficult to complete due to logistical and socioeconomic constraints [21,22]. Only 35.6% of the caregivers participated in community-based malaria vaccination programs, reflecting a concerningly low level of public engagement. Community participation is essential for achieving herd immunity and sustaining immunization coverage in high-transmission settings [23].

This study found that only 24.1% of caregivers demonstrated good overall practices regarding the malaria vaccine, reflecting a generally low level of engagement and adherence to vaccination guidelines in the Tiko Health District. Such poor practices may be explained by limited health education, inadequate awareness of vaccination schedules, and low community sensitization during the initial rollout of the RTS,S/AS01 malaria vaccine [24,25]. Similar findings of suboptimal

caregiver practices and uptake have been reported in some sub-Saharan African countries like Ghana and Kenya, where insufficient caregiver knowledge and low confidence in the vaccine affected uptake despite availability [26,27]. Evidence from Cameroon has also highlighted the role of poor communication strategies and weak community mobilization in hindering vaccine acceptance and adherence [28]. These findings underscore the need for intensified health education, sustained community sensitization, and integration of malaria vaccine promotion into routine immunization services.

This study identified several variables that were independently associated with malaria vaccine uptake among children aged 0–5 years in the Tiko Health District, which were; sex, profession, relationship with child, income, accessibility of health services, history of malaria infection, source of information, and trust in health workers. Children whose primary caregivers were female were four times more likely to be vaccinated compared to those cared for by males. This confirms findings from previous studies indicating that mothers are typically more involved in child health decisions and are more likely to engage in preventive healthcare measures such as vaccination [19]. In many African settings, traditional gender roles place child health responsibilities largely on women, who often accompany children to health facilities [20]. Caregivers working in health-related professions were three times more likely to vaccinate their children. This is could due to the fact that, those with medical or health training are more likely to understand the benefits of vaccines, perceive their safety positively, and access services more readily [21]. This aligns with WHO's assertion that health literacy among caregivers significantly improves vaccine confidence and completion [1]. Children living with their biological parents were significantly more likely to be vaccinated than those under the care of other relatives or unrelated guardians. This finding is supported by research in Tanzania and Kenya showing that non-parental caregivers often have lower prioritization of child health, less consistent healthcare-seeking behavior, and reduced vaccination rates [22].

In addition, higher household income was positively associated with vaccine uptake. Children from households with a monthly income between 89USD and 179USD were more than twice more likely to be vaccinated compared to those from households earning less than 89USD. This is similar to a study carried out in Cameroon, where economic stability was found to facilitates better access to healthcare services and adherence to vaccination schedules [23]. Caregivers who reported easy access to health services were about twice more likely to have their children vaccinated. Accessibility plays a crucial role in healthcare utilization, and efforts to improve healthcare infrastructure and reduce travel barriers are essential for increasing vaccine coverage [29]. Caregivers who received information about the malaria vaccine from healthcare providers were more likely to vaccinate their children. Trusted sources of information, such as healthcare workers, play a vital role in influencing health behaviors. In Cameroon, community-based risk communication involving health workers has been effective in promoting vaccine acceptance [29]. Trust in healthcare workers was a strong predictor of vaccine uptake. Caregivers who trusted health workers were about six times more likely to vaccinate their children, building and maintaining trust through consistent, transparent, and culturally sensitive communication is essential for the success of vaccination programs [29].

To improve caregiver practices toward malaria vaccination, targeted community sensitization and health education campaigns should be implemented to increase awareness of vaccine schedules, benefits, and the importance of completing all required doses. Efforts should focus on both female and male caregivers to promote shared responsibility in child healthcare [27,30].

To address the determinants associated with malaria vaccine uptake, strategies should be put in place to improve access to vaccination services, particularly for low-income households and those in hard-to-reach areas. Strengthening trust in healthcare workers through community engagement and training in effective communication is also essential [31].

## Strengths and limitations

This study has several strengths. It employed a community-based design with a relatively large sample size, enhancing the generalizability of the findings within the Tiko Health District. The use of a structured questionnaire and trained research assistants ensured standardized data collection and improved data quality. Additionally, the study identified

a range of caregiver and household factors influencing malaria vaccine uptake, providing actionable insights for public health interventions.

However, the study also has limitations. Its cross-sectional design limits the ability to infer causality between identified factors and vaccine uptake. Self-reported vaccination status may be subject to recall bias, although the verification of vaccination records for most participants helped mitigate this risk. Finally, the study was conducted in a single health district, which may limit the generalizability of the findings to other regions in Cameroon with different sociocultural or health service contexts.

## Conclusion

The uptake of the malaria vaccine among under-five children in the Tiko Health District was low, and overall good caregiver practice was also limited. Significant predictors of vaccine uptake included caregiver sex, health profession, relationship to the child, household income, healthcare accessibility, trust in health workers, and source of information. Conversely, a history of malaria infection in the child was negatively associated with uptake. Public health interventions aimed at improving vaccine coverage should prioritize education and sensitization campaigns, improve healthcare service accessibility, and build community trust in health professionals. These actions will be essential for the success of malaria vaccination rollout in similar contexts.

## Supporting information

**S1 Data. Dataset containing anonymized data used in this study.**
(XLSX)

## Author contributions

**Conceptualization:** Idang Maureen Abiache, Divine Nsobinenyui, Chrisantus Eweh Ukah, Yunika Larissa Kumenyuy, Ngu Claudia Ngeha, Randolf Wefuan, Syveline Zuh Dang, Ndip Esther Ndip, Mirabelle Pandong Feguem, Dickson S. Nsagha.

**Data curation:** Divine Nsobinenyui, Chrisantus Eweh Ukah, Yunika Larissa Kumenyuy, Ndip Esther Ndip, Mirabelle Pandong Feguem, Dickson S. Nsagha.

**Formal analysis:** Chrisantus Eweh Ukah, Yunika Larissa Kumenyuy, Randolf Wefuan, Ndip Esther Ndip.

**Investigation:** Idang Maureen Abiache, Divine Nsobinenyui, Chrisantus Eweh Ukah, Yunika Larissa Kumenyuy, Ngu Claudia Ngeha, Randolf Wefuan, Syveline Zuh Dang, Ndip Esther Ndip, Mirabelle Pandong Feguem, Dickson S. Nsagha.

**Methodology:** Idang Maureen Abiache, Divine Nsobinenyui, Chrisantus Eweh Ukah, Yunika Larissa Kumenyuy, Randolf Wefuan, Syveline Zuh Dang, Ndip Esther Ndip, Mirabelle Pandong Feguem, Dickson S. Nsagha.

**Project administration:** Idang Maureen Abiache, Divine Nsobinenyui, Chrisantus Eweh Ukah, Yunika Larissa Kumenyuy, Ndip Esther Ndip, Dickson S. Nsagha.

**Resources:** Chrisantus Eweh Ukah.

**Supervision:** Divine Nsobinenyui, Chrisantus Eweh Ukah, Dickson S. Nsagha.

**Validation:** Divine Nsobinenyui, Chrisantus Eweh Ukah, Ndip Esther Ndip, Dickson S. Nsagha.

**Visualization:** Idang Maureen Abiache, Chrisantus Eweh Ukah, Yunika Larissa Kumenyuy, Randolf Wefuan, Ndip Esther Ndip.

**Writing – original draft:** Idang Maureen Abiache, Divine Nsobinenyui, Chrisantus Eweh Ukah, Yunika Larissa Kumenyuy, Ngu Claudia Ngeha, Randolf Wefuan, Syveline Zuh Dang, Ndip Esther Ndip, Mirabelle Pandong Feguem, Dickson S. Nsagha.

**Writing – review & editing:** Idang Maureen Abiache, Divine Nsobinenyui, Chrisantus Eweh Ukah, Yunika Larissa Kumenyuy, Ngu Claudia Ngeha, Randolf Wefuan, Syveline Zuh Dang, Ndip Esther Ndip, Mirabelle Pandong Feguem, Dickson S. Nsagha.

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
