## [Decision Letter · Decision Letter 0]

11 Jun 2025

PGPH-D-25-01162

Uptake and determinants of malaria vaccination among under-five children in the Tiko Health District: a community based cross-sectional study

Dear Dr. Ukah,

Thank you for submitting your manuscript to PLOS Global Public Health. After careful consideration, we feel that it has merit but does not fully meet PLOS Global Public Health’s publication criteria as it currently stands. Therefore, we invite you to submit a revised version of the manuscript that addresses the points raised during the review process.

We look forward to receiving your revised manuscript.

Kind regards,

Abhinav Sinha, M.D.

Academic Editor

Journal Requirements:

Additional Editor Comments (if provided):

Reviewers' comments:

Reviewer's Responses to Questions

**Comments to the Author**

1. Does this manuscript meet PLOS Global Public Health’s publication criteria ? Is the manuscript technically sound, and do the data support the conclusions? The manuscript must describe methodologically and ethically rigorous research with conclusions that are appropriately drawn based on the data presented.

Reviewer #1: Partly

Reviewer #2: Partly

2. Has the statistical analysis been performed appropriately and rigorously?

Reviewer #1: Yes

Reviewer #2: Yes

3. Have the authors made all data underlying the findings in their manuscript fully available (please refer to the Data Availability Statement at the start of the manuscript PDF file)?

Reviewer #1: Yes

Reviewer #2: Yes

4. Is the manuscript presented in an intelligible fashion and written in standard English?

Reviewer #1: No

Reviewer #2: No

5. Review Comments to the Author

Reviewer #1: the manuscript is fairly presented , with the quality of the data ,it can be further improvised , the title , method and conclusion needs to be aligned to make the work more acceptable . The statistical analysis done needs to be defined properly - specially the regression - detailing the selection of variable , how the confounders were adjusted and the reference in categorical variable . The intervention group and the conclusion needs to be defined properly.

Reviewer #2: The study has evaluated the factors affecting the malaria vaccine uptake among under-five children by studying the socio-economic status and vaccine-related knowledge of caregivers.

The authors have addressed an important aspect related to malaria vaccine uptake; however, overall, the manuscript is casually written with grammatical errors at many places. I would recommend the authors to carefully revise the manuscript in-line of the below-mentioned comments. I hope the comments would help in improving the quality of manuscript according to the journal standard.

Comments:    

Line 65-67: Please provide the recents numbers from WHO World Malaria Report-2024.Line 111: Sample size formula has mentioned d2 in the denominator, however, E2 is abbreviated. Line 118-123: Please provide the detailed methodology adopted for- Simple random sampling- Out of eight health areas, how many were selected and what strategy was used i.e. computer generated random numbers or using random number table or lottery method etc. Briefly mention about the PPS allocation- what was the proportion of the population that each health area represents? What was the number of caregivers from each of the health areas? Systematic sampling was used to select eligible households- An estimated total number of households in each health area may be given, of which how many households were selected and how they were selected i.e. skipping any fixed number of houses until reaching the final sample size or any other method?Line 162: A vast majority 329 (80.2%) of the caregivers were direct parents of the under......children? Please correct.

Table 1: Household income- Please mention currency value.Smoking status- Mention smoker/ non-smoker instead of smoke/ not smokeChild ever attacked with malaria- Replace it with "Child ever had Malaria Episodes"Line 168-169: Sentence incomplete. Recheck it.Line 169-170: Rephrase the sentences and check for correct grammar in the sentences.

Table 2: Ensures that child complete recommended vaccine doses- Rephrase to "Compliance  of recommended vaccine doses by the child".you actively participate in community vaccination programs that offer the malaria vaccine to children- Is this information related to community vaccination programs or about malaria vaccination being done under community vaccination? Rephrase the sentence accordingly.I seek medical advice when child experiences negative symptoms after vaccination- Rephrase it to "Seeks medical advice in case of negative symptoms after vaccination". Figure 1: Mention malaria vaccine instead of just vaccination.

Line 191-193: Check for grammar.Table 3: 

COR- Abbreviate. Also, this table estimated the odds ratios of the factors associated with vaccine uptake. Therefore, No vaccination column is not required. For eg. Sex- Female (Give total number)-Child vaccinated (number (117), percentage (28.5%).     Replace- Drink alcohol to Alcohol consumption.Child has suffered from malaria before- Replace with "History of malaria in child"Go for general vaccination- Replace with "undergoes standard immunization"Line 206, 208 and thereafter: adjusted odds ratio was mentioned, however, respective table 4 shows COR? Also, mention about the factors which were adjusted to calculate the aOR in the method section. Line 211: XAF- abbreviate Line 218-220: Sentence incomplete

Table 4:COR- Abbreviate. Also, this table estimated the odds ratios of the factors associated with vaccine uptake. Therefore, No vaccination column is not required.

Line 230-231: correct grammar in the sentence.Line 233-235: rephrase the sentence for better clarity.

6. PLOS authors have the option to publish the peer review history of their article (what does this mean? ). If published, this will include your full peer review and any attached files.

**Do you want your identity to be public for this peer review?** For information about this choice, including consent withdrawal, please see our Privacy Policy .

Reviewer #1: No

Reviewer #2: **Yes: ** Hari Shankar

---

## [Decision Letter · Decision Letter 1]

7 Sep 2025

PGPH-D-25-01162R1

Caregivers’ practices and factors associated with malaria vaccine uptake among under-five children in the Tiko Health District, Cameroon: a community based cross-sectional study

Dear Dr. Ukah,

Thank you for submitting your manuscript to PLOS Global Public Health. After careful consideration, we feel that it has merit but does not fully meet PLOS Global Public Health’s publication criteria as it currently stands. Therefore, we invite you to submit a revised version of the manuscript that addresses the points raised during the review process.

We look forward to receiving your revised manuscript.

Kind regards,

David Musoke, PhD

Academic Editor

Journal Requirements:

Additional Editor Comments (if provided):

The title of the manuscript is not consistent with the aim of the study in the abstract.

Line 41: Structured questionnaires or rather a restructured questionnaire?

Being an international journal, it is recommended that the currency should be converted to USD in the entire manuscript.

Line 92: … study aimed to …

Lines 119 and 120: Why sampling technique twice?

What was the background and level of training of the Research Assistants?

More details on the training of research assistants including content and number of days are needed.

How was the questionnaire designed? Was reference made to any existing literature? If so, the literature should be cited.

How were ‘caregivers’ defined? Did they include mothers? These details seem lacking.

Figure 2 may not be necessary.

Statistically significant results should be indicated in the tables.

The first paragraph of the discussion should provide an overview of the findings and their general significance to public health. Detailed discussion of the findings should begin in the second paragraph of the discussion.

Separate paragraphs for various key findings are needed in the discussion. Currently, the discussion is a single long paragraph.

Study limitations (and any strengths) are needed at the end of the discussion.

A subsection of ‘recommendations’ is usually not expected in a manuscript.

No need to have statistics in the conclusion.

Reviewers' comments:

Reviewer's Responses to Questions

**Comments to the Author**

1. If the authors have adequately addressed your comments raised in a previous round of review and you feel that this manuscript is now acceptable for publication, you may indicate that here to bypass the “Comments to the Author” section, enter your conflict of interest statement in the “Confidential to Editor” section, and submit your "Accept" recommendation.

Reviewer #1: All comments have been addressed

2. Does this manuscript meet PLOS Global Public Health’s publication criteria ? Is the manuscript technically sound, and do the data support the conclusions? The manuscript must describe methodologically and ethically rigorous research with conclusions that are appropriately drawn based on the data presented.

Reviewer #1: Yes

3. Has the statistical analysis been performed appropriately and rigorously?

Reviewer #1: Yes

4. Have the authors made all data underlying the findings in their manuscript fully available (please refer to the Data Availability Statement at the start of the manuscript PDF file)?

Reviewer #1: Yes

5. Is the manuscript presented in an intelligible fashion and written in standard English?

Reviewer #1: Yes

6. Review Comments to the Author

Reviewer #1: The modified version is much improved.

7. PLOS authors have the option to publish the peer review history of their article (what does this mean? ). If published, this will include your full peer review and any attached files.

**Do you want your identity to be public for this peer review?** For information about this choice, including consent withdrawal, please see our Privacy Policy .

Reviewer #1: **Yes: ** Dr Shrinivasa B Marinaik

---

## [Editor Report · Decision Letter 2]

18 Sep 2025

PGPH-D-25-01162R2

Caregivers’ practices and factors associated with malaria vaccine uptake among under-five children in the Tiko Health District, Cameroon: a community based cross-sectional study

Dear Dr. Ukah,

Thank you for submitting your manuscript to PLOS Global Public Health. After careful consideration, we feel that it has merit but does not fully meet PLOS Global Public Health’s publication criteria as it currently stands. Therefore, we invite you to submit a revised version of the manuscript that addresses the points raised during the review process.

We look forward to receiving your revised manuscript.

Kind regards,

David Musoke, PhD

Academic Editor

Journal Requirements:

Additional Editor Comments (if provided):

'Questionnaires' is still used in the abstract, line 344 and elsewhere.

Lines 275 to 277 may be added to the first paragraph of the discussion.

Check the font type for lines 186 to 188.

Line 204: check punctuation (full stop).

The amounts in US dollars in the tables may be rounded off to the nearest whole number.

Line 293 to 298: No reference to existing literature is made. Indeed, it is a shallow discussion paragraph.

Very long discussion paragraph from line 299 to 332.

Line 333 to 340: reference to existing literature is needed for all discussion paragraphs.

Conclusion may be a single paragraph.

Reviewers' comments:

Figure Resubmissions:

---

## [Editor Report · Decision Letter 3]

6 Oct 2025

Caregivers’ practices and factors associated with malaria vaccine uptake among under-five children in the Tiko Health District, Cameroon: a community based cross-sectional study

PGPH-D-25-01162R3

Dear Dr. Ukah,

We are pleased to inform you that your manuscript 'Caregivers’ practices and factors associated with malaria vaccine uptake among under-five children in the Tiko Health District, Cameroon: a community based cross-sectional study' has been provisionally accepted for publication in PLOS Global Public Health.

Best regards,

David Musoke, PhD

Academic Editor